# Estimated contribution of most commonly consumed industrialized processed foods to salt intake and iodine intakes in Sri Lanka

Renuka Jayatissa[1]◉*, Chandima Haturusinghe[1]◉, Jacky Knowles[2]◉, Karen Codling[2]‡, Jonathan Gorstein[2]‡

1 Department of Nutrition, Medical Research Institute, Colombo, Sri Lanka, 2 Iodine Global Network, Ontario, Canada

◉ These authors contributed equally to this work.
‡ KC and JG also contributed equally to this work.
* renukajayatissa@ymail.com

**Data Availability Statement:** All relevant data are within the manuscript.

## Abstract

In Sri Lanka dietary patterns are shifting towards increased consumption of industrially processed foods (IPF). This study aimed to estimate the contribution of IPF to salt and iodine intake and assess the possible impact of salt reduction on iodized salt intake. The assessment was conducted using guidance published by the Iodine Global Network. National nutrition and household income expenditure surveys were used to estimate adult per capita consumption of household salt and commonly consumed salt-containing IPF. Industry and laboratory data were used to quantify salt content of IPF. Modelling estimated the potential and current iodine intake from consumption of household salt and using iodized salt in the identified IPF. Estimates were adjusted to investigate the likely impact on iodine intake of achieving 30% salt reduction. IPF included were bread, dried fish and biscuits, with daily per capita consumption of 32g, 10g and 7g respectively. Daily intake of household salt was estimated to be 8.5g. Potential average national daily iodine intake if all salt in these products was iodized was 166μg. Estimated current daily iodine intake, based on iodization of 78% of household salt and dried fish being made with non-iodized salt, was 111μg nationally, ranging from 90 to 145μg provincially. Estimated potential and current iodine intakes were above the estimated average requirement of 95μg iodine for adults, however, current intake was below the recommended nutrient intake of 150μg. If the 30% salt reduction target is achieved, estimated current iodine intake from household salt, bread and biscuits could decrease to 78μg. The assessment together with data for iodine status suggest that current iodine intake of adults in Sri Lanka is adequate. Recommendations to sustain with reduced salt intake are to strengthen monitoring of population iodine status and of food industry use of iodized salt, and to adjust the salt iodine levels if needed.

**Funding:** Initials of the authors who received each award - RJ Grant numbers awarded to each author - Investment ID OPP1195090 The full name of each funder - Bill and Melinda Gate Foundation URL of each funder website - None Did the sponsors or funders play any role in the study design, data collection and analysis, decision to publish, or preparation of the manuscript? No The funders had no role in study design, data collection and analysis, decision to publish, or preparation of the manuscript.

**Competing interests:** No authors have competing interests.

## Introduction

The World Health Organization (WHO) has established a global target of a 30% relative reduction in mean population intake of salt/sodium, towards a recommended level of less than 5g salt per day for adults. This was one of nine voluntary global targets to achieve the goal of reducing the preventable and avoidable burden of morbidity, mortality and disability due to noncommunicable diseases (NCDs) by means of multisectoral collaboration and cooperation at national, regional and global level, by 2025 [1].

A salt reduction strategy was implemented in Sri Lanka in 2019 by the Ministry of Health, with a comprehensive action plan to reduce salt intake among adults to 8g or less per day by 2025 [2]. The national action plan includes a mandatory activity to reduce the salt content of commonly consumed Industrially Processed Foods (IPF), such as salted snacks, through product reformulation [2]. Programs to achieve optimal iodine intake rely on iodized salt to ensure sufficient iodine intake for the population. Therefore, there is an urgent need for closely coordinated programs to reduce dietary salt intake while sustaining iodine intake to maximize benefit to the public [3].

In Sri Lanka, universal salt iodization became mandatory in 1995 [4] and has resulted in sustained control of iodine deficiency disorders over the past 15 years. Surveys during this period have consistently shown that over 95% of households were using salt with at least some iodine (>5 mg/kg). In 2016, 78% of households were using salt with 15 mg/kg iodine or above and the median salt iodine content across all households was 21mg/kg, with interprovincial differences ranging from 18 to 28 mg/kg [5, 6]. The same 2016 survey included assessment of iodine status among 5,000 children 6 to 12 years of age and indicated adequate iodine status among this group nationally, median urinary iodine concentration (MUIC) 233 µg/L, and across the provinces, MUIC ranged from179 µg/L in Uva province to 297 µg/L in the Northern province. A 2015 survey among 962 pregnant women nationally reported a MUIC of 158 µg/L, indicating sufficient status among this group also, although MUIC was found to be slightly lower than optimal during the first trimester [7].

Recognizing that only 78% of households were using adequately iodized salt, these studies showing adequate iodine status suggest a contribution to iodine intake from sources other than only household salt [6]. Food industry salt has the potential to be another significant source of dietary iodine if iodized and it was, therefore, considered necessary to assess the contribution of iodine from iodized salt in processed food. In Sri Lanka, as globally, food consumption patterns and trends are shifting towards increased consumption of processed foods the 2015 WHO STEPS survey reported that 27% of respondents responded that they always or often eat processed food and that on average 1.4 meals per week eaten outside the home [8].

Achievement of optimal iodine nutrition can be strengthened and more effectively managed if the program is based on data about all major dietary sources of iodized salt together with data on iodine status. However, no systematic assessment has been conducted so far in Sri Lanka. This information would help strengthen monitoring and enforcement processes to ensure all food grade salt is iodized, and to subsequently adjust salt iodization standards as necessary to sustain optimal iodine intake, while preventing both deficiency and excessive consumption [9].

In response to a growing awareness that IPF salt was an increasingly important source of dietary salt globally, yet the presence and level of iodine in salt used by the food industry is often not known, the Iodine Global Network (IGN) developed draft programme guidance on "The use of iodized salt in industrially processed food" [10]. A team from Sri Lanka responded to the request for expressions of interest to pilot this draft programme guidance, with technical support from IGN. The aim was to identify the potential and estimated current iodine intake

through the use of iodized salt in processed food manufacture and use the resulting data to further help include large-scale food industry partners in consolidated efforts to ensure iodization of all food grade salt. Hence this study was conducted with the following objectives: to identify commonly consumed processed foods contributing to salt intake across the population, to estimate per capita salt intake from industrailly produced forms of these foods and the percent of food industry salt that is iodized, and to analyze the potential contribution of iodized salt used in the manufacture of those food to adult recommended daily nutrient reference values for iodine.

## Method

Nationally representative relevant data sources were identified, in accordance with the methodology of the IGN programme guidance [10]. Salt containing IPF were defined in the IGN guidance as products from food industries that purchase salt in bulk and produce foods with relatively wide market reach. Products are usually packaged and branded [10].

### Data sources used for the assessment

**a. National Nutrition and Micronutrient Survey of Pregnant Women 2015 (NNMSPW 2015) and National Nutrition Survey of Lactating Mothers 2015 (NNSLM 2015) [7, 11].** Survey participants for these two surveys were lactating mothers of infants 0 to 6 months of age, or pregnant women who were registered with a public health midwife. The surveys were cluster surveys stratified by district. Within each of the 25 districts, 30 sampling areas were selected from the 2011 census data using probability proportional to size sampling. Ten lactating and 10 pregnant women were then selected using systematic random sampling, from the birth and immunization register (lactating women) and the pregnant mother's registry, in the 30 sampling units in each district. These surveys reported data on the frequency of consumption (mean number of days in the week before the interview) of certain salt containing processed food and the estimated quantity of salt used by the household monthly. The salt containing processed foods included in the survey were dried fish, bread, canned fish, margarine, cheese, instant noodles, biscuits, cakes, cereals and butter. Quantity of consumption data were not available and there was no information on whether the foods were homemade or industrially processed. The selection of processed foods to include in the final assessment took into account estimates of intake from the Household Income and Expenditure Survey (see below) and the low frequency of consumption (more than once per week) for canned fish, margarine, cheese and butter. Per capita daily salt consumption was estimated by dividing the reported monthly quantity of salt used by the number of household members then dividing by 30 days. District level data for salt consumption were proportionately weighted for the district population and weighted average was taken to determine the provincial estimates.

**b. Household Income Expenditure Survey (HIES) 2016 [12].** This was a household survey with 2,500 primary sampling units (2011 census blocks) selected from across all 25 districts, with a selection probability based on an updated listing of the number of housing units available in the census block. Ten housing units were selected using random systematic sampling from each of the selected census blocks. The final sample size was 21,756 households. Supplementary HIES tables included estimates for per capita monthly consumption of salt containing processed foods such as dried fish, bread, canned fish, margarine, cheese, biscuits, cakes, and butter. The HIES methodology reported calculating per capita consumption by dividing household consumption by the number of household consumption by household members. Data to calculate adult male equivalents was not available. The intake of salt from calculated daily per capita consumption quantities of each of these foods was estimated along

with the corresponding potential iodine intake, assuming salt was iodized to the national standard of 22.5 mg/kg. Out of 10 top processed foods consumed, daily per capita intake of iodine from the use of iodized salt in each product was estimated to be below 1.5 µg for all foods except for bread, dried fish and biscuits. Based on this assessment and the low frequency of consumption of certain foods reported above, bread, dried fish and biscuits were the only foods reported above, included in the final analysis. Consumption estimates were available by province for household salt and bread, but not for biscuits and dried fish. Therefore, for provincial level modelling of biscuits and dried fish intake, it was assumed there was the same level of consumption at national and provincial level (see the limitations of the study in the discussion section).

**c. Food industry data.** The salt content of bread and biscuits was determined based on information from the food industry (two out of three national biscuits producers and the All-Ceylon Bakery Owners Association). Estimated average percent salt content by product weight of bread and biscuits were 1.0% and 1.4% respectively.

**d. Laboratory data on salt content of dried fish.** Information extracted from national laboratory surveillance database established by the Medical Research Institute for salt content of commonly consumed food. The salt content of 10 commonly consumed varieties of dried fish was obtained, using an average of measured salt content from 10 samples from each variety of fish. The average salt content was 16%, which is in line with findings from an analysis of commercial dried fish salt content conducted in Chennai, India [13].

## Modelling

The estimate for daily per capita salt intake from each selected IPF was done by multiplying the percent salt content by the estimated daily consumption quantity in grams (Eq 1).

$$\begin{aligned} &\textit{Estimated daily salt intake from each product (g)} \\ &= \textit{Estimated daily per capita consumption of each product (g)} \quad\quad \text{(Eq 1)} \\ &\textit{x product salt content (\% product weight)} \end{aligned}$$

**Potential iodine intake** from iodized household salt and salt used in IPF production was based on: 100% of salt being iodized at the mean of the national standard level (22.5 mg/kg iodine, based on the national standard of 15–30 mg/kg) [4], and an estimate for iodine loss of 30 percent in these products from the production to consumption (0.7 in Eq 1). Losses can vary widely and depend on the salt iodization process, the quality of salt and packaging materials, the climatic conditions, and the food industry process. Thirty percent loss is the estimate used by WHO in their guideline on salt iodization [14]. This calculation is shown in Eq 2.

$$\begin{aligned} &\textit{Potential iodine intake (µg) from daily intake of household salt and IPF salt} \\ &= \textit{Estimated daily intake from each product (g) (see Eq 1) x 22.5 x 0.7} \end{aligned} \quad \text{(Eq 2)}$$

Estimates for **current iodine intake** from salt in these foods was calculated using the best available current data for the percent iodized salt, which was that dried fish is made with non-iodized salt, bread and biscuits are likely to be made with iodized salt only and 78% of households were using salt with adequate iodine levels, assumed from this calculation to be 22.5 mg/kg iodine [6] shown in Eq 3.

$$\begin{aligned} &\textit{Current estimated iodine intake (µg) from daily intake of household salt and IPF salt} \\ &= \textit{Estimated daily salt intake from each province (g) (see Eq 1) x22.5 x 0.7 x a} \quad \text{(Eq 3)} \\ &\textit{Where } a = 0 \textit{ for dried fish}, 1 \textit{ for bread and biscuit}, \textit{and } 0.78 \textit{ for household salt}. \end{aligned}$$

Potential and current iodine intake from iodized salt was also calculated for a scenario where there was 30% reduction in salt intake from household salt and all IPF. This was calculated using the outcome of Eq 3 multiplied by 0.7.

The estimates for daily iodine intake from household salt and salt in IPF were compared with Estimated Average Requirements (EAR), Recommended Nutrient Intake (RNI) and Tolerable Upper Level (UL) for iodine in adults (non-pregnant) of 95, 150 and 600 μg respectively [15] and are presented as percent contribution to these nutrient reference values. Nutrient reference values vary by age group and pregnancy status. Non-pregnant adult reference values were used as the best approximation, given that estimates for typical intake of selected foods are per household member.

## Results

In the NNMSPW 2015, the median salt intake per household member was 8.9g, in the NNSLM 2015, the median salt intake per household member was 5.3g and the mean was 8.5g. HIES data reported an approximate mean salt intake of 6.9g. An approximate estimate of 8.5g household salt intake per household member per day was therefore used for the assessment, based on all these sources of data and an estimate of 11.4 g salt intake from a 24-hr urine excretion study [16].

Table 1 shows average daily intake per household member at the national and provincial levels for bread, average national intake of 32g/day; and at the national level only for dried fish, 10g/day; and biscuits, 7g/day (provincial estimates were not available). Bread consumption varied between provinces, the average daily per capita consumption of bread was highest (45g) in the Western province while the second highest consumption was in the Northern province (43g). The lowest was in the Uva province (15g).

Table 2 shows the average daily per capita intake of salt from consumption of household salt and selected IPF. Total average daily per capita intake from all 4 products was approximately 10.5g at the national level. It varied from 9.2g in Western, Central, Uva and Sabaragamuwa provinces to 11.7g/day in Eastern province. The major contribution to daily salt intake came from household salt (8.5g) and dried fish (1.6g). When compared with the WHO daily recommended maximum salt intake of 5g for adults, salt intake from the 4 foods reported here contributes to more than double the recommended maximum.

**Table 1. Average daily consumption of commonly consumed salt-containing IPF per household member, nationally and by province.**

| Province | Average daily consumption of selected IPF (g) per household member | | |
|---|---|---|---|
| | **Bread** | **Dried fish**[a] | **Biscuits**[a] |
| Western | 45 | 10 | 7 |
| Southern | 28 | 10 | 7 |
| Central | 30 | 10 | 7 |
| Northern | 43 | 10 | 7 |
| Eastern | 36 | 10 | 7 |
| North Western | 24 | 10 | 7 |
| North Central | 18 | 10 | 7 |
| Uva | 15 | 10 | 7 |
| Sabaragamuwa | 21 | 10 | 7 |
| **Sri Lanka** | **32** | **10** | **7** |

[a] Provincial intake estimates based on the national estimate in the absence of province-specific data for these products.

**Table 2. Average daily salt intake per household member from household salt and selected commonly consumed IPF, nationally and by province.**

| Province | Average daily salt intake per household member (g) | | | | |
|---|---|---|---|---|---|
| | Household Salt | Bread | Dried fish | Biscuits | Total |
| Western | 7.1 | 0.5 | 1.6 | 0.1 | **9.2** |
| Southern | 8.3 | 0.3 | 1.6 | 0.1 | **10.3** |
| Central | 7.2 | 0.3 | 1.6 | 0.1 | **9.2** |
| Northern | 9.3 | 0.4 | 1.6 | 0.1 | **11.4** |
| Eastern | 9.6 | 0.4 | 1.6 | 0.1 | **11.7** |
| North Western | 7.7 | 0.2 | 1.6 | 0.1 | **9.6** |
| North Central | 8.4 | 0.2 | 1.6 | 0.1 | **10.3** |
| Uva | 7.4 | 0.2 | 1.6 | 0.1 | **9.2** |
| Sabaragamuwa | 7.3 | 0.2 | 1.6 | 0.1 | **9.2** |
| **Sri Lanka** | **8.5** | **0.3** | **1.6** | **0.1** | 10.5 |
| % WHO recommended max salt intake of 5g/day for adults | 170% | 6% | 32% | 2% | 210% |

Table 3 presents the potential (166 μg nationally) and estimated current (111 μg nationally) daily iodine intake from iodized household salt and iodized salt in the 3 selected IPF. Relative potential iodine intake from each product reflects salt intake (Table 2) and, as such, the major share of potential iodine intake in all provinces is from household salt (134 μg) then from dried fish (25 μg). However, estimated current iodine intake from household salt is 104 μg, based on 78% of household salt being iodized, and from salt in dried fish is 0 μg since it is not made with iodized salt. The relatively low volume of consumption of bread and biscuits resulted in a low potential and estimated current iodine intake from these products which are, together, estimated to contribute to 7 μg iodine per day.

Fig 1 illustrates that iodized household salt and the use of iodized salt in selected IPF have the potential to fulfil 174% of the EAR for iodine (95 μg) and 110% of the RNI for iodine (150 μg). The estimated current iodine intake (based on estimates of actual percent salt

**Table 3. Estimated potential (P) and current (C) daily iodine intake (μg) from household salt and commonly consumed IPF in non-pregnant adults.**

| Province | Current estimates for % household salt with at least 15 mg/kg iodine | Average potential daily iodine intake (μg) (P)—if salt in all 4 products is iodized and estimated current daily iodine intake (μg) (C)—based on current levels of salt iodization | | | | | | | | |
|---|---|---|---|---|---|---|---|---|---|---|
| | | Household Salt | | Bread | | Dried fish | | Biscuits | | Total | |
| | | P | C | P | C | P | C | P | C | P | C |
| Western | 78% | 112 | 87 | 7 | 7 | 25 | 0 | 2 | 2 | 146 | 96 |
| Southern | 71% | 131 | 93 | 4 | 4 | 25 | 0 | 2 | 2 | 162 | 99 |
| Central | 90% | 113 | 102 | 5 | 5 | 25 | 0 | 2 | 2 | 145 | 108 |
| Northern | 83% | 146 | 122 | 7 | 7 | 25 | 0 | 2 | 2 | 180 | 130 |
| Eastern | 91% | 151 | 138 | 6 | 6 | 25 | 0 | 2 | 2 | 184 | 145 |
| North Western | 70% | 121 | 85 | 4 | 4 | 25 | 0 | 2 | 2 | 152 | 90 |
| North Central | 68% | 132 | 90 | 3 | 3 | 25 | 0 | 2 | 2 | 162 | 94 |
| Uva | 80% | 117 | 93 | 2 | 2 | 25 | 0 | 2 | 2 | 146 | 97 |
| Sabaragamuwa | 82% | 115 | 94 | 3 | 3 | 25 | 0 | 2 | 2 | 145 | 99 |
| **Sri Lanka** | **78%** | **134** | **104** | **5** | **5** | **25** | **0** | **2** | **2** | **166** | **111** |

C - Estimated current intake is based on 78% of household salt being iodized to at least 15 mg/kg, 100% iodized salt used in bakeries and by biscuit manufacturers and non-iodized salt used for dried fish production.

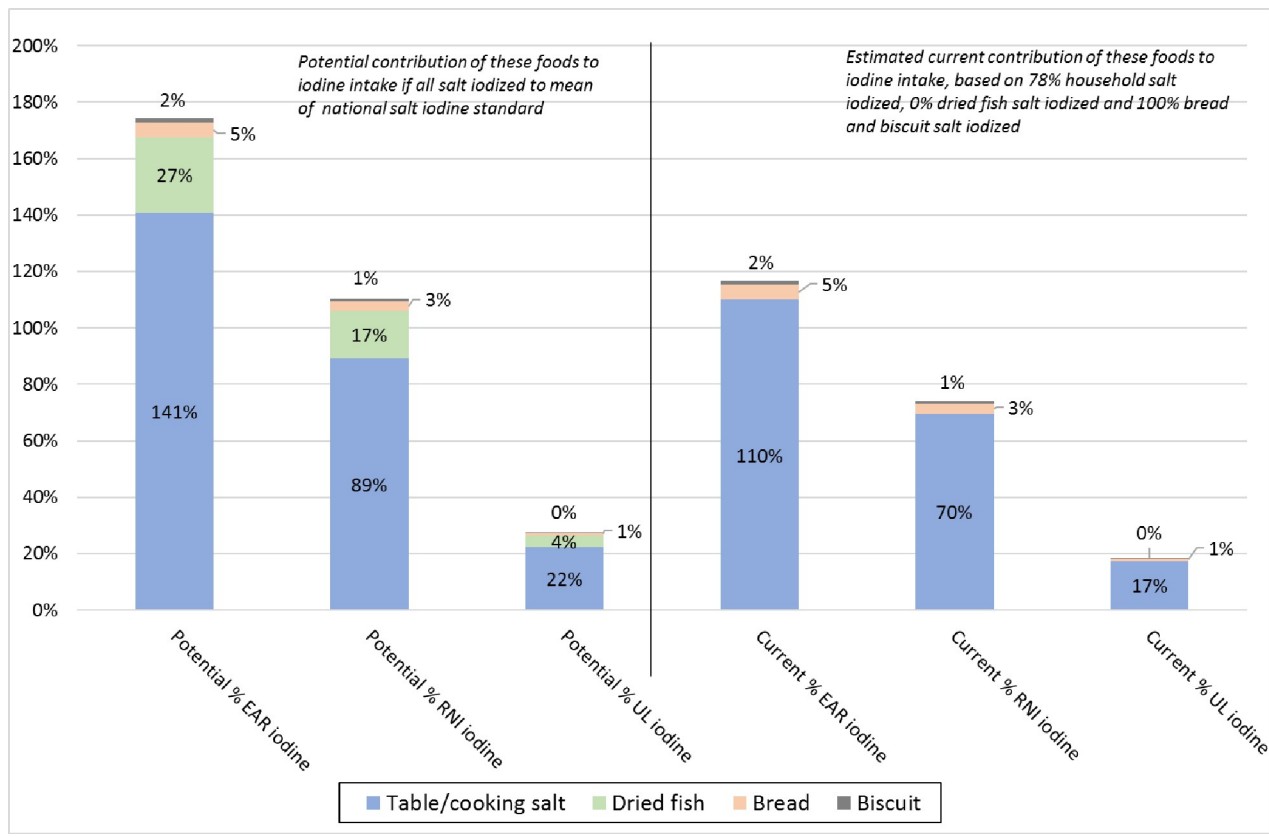

**Fig 1. Contribution of iodized household salt and in IPF to potential (all salt iodized) and estimated current (based on salt iodization practices) iodine levels, in relation to nutrient reference values for iodine for non-pregnant adults.** [EAR = Estimated Average Requirement; RNI = Recommended Nutrient intake; UL = Upper Tolerable Intake for iodine. The values for these parameters for non-pregnant adults are 95, 150 and 600 µg iodine, respectively].

iodized) of 111 µg iodine from household salt, bread and biscuits would be above the EAR for iodine and contribute to 74% of the RNI for iodine. Both potential and estimated current iodine intake from household salt and the IPF are well below the UL for iodine of 600 µg, at 28% and 19% of the UL respectively. At the provincial level, estimated current daily iodine intake from household salt and the three IPF is estimated to range from 90 µg (94% of the EAR and 60% of the RNI) in the North Western province to 145 µg (153% of the EAR and 97% of the RNI) in the Eastern province.

Fig 2. illustrates the possible impact on iodine intake if the salt reduction strategy (30 percent reduction target) is successfully implemented and is assumed to apply across all products evenly. It would reduce the contribution to daily iodine intake through iodized salt in these foods to 78 µg nationally (82% of the EAR for iodine and 52% of the RNI for iodine, for adults). This level of iodine is far below, 13% of, the UL for iodine of 600 µg.

The relative possible effect on iodine intake from salt reduction by province shown in Fig 2 is in line with the differences in estimated current iodine intake shown in Table 3. If a 30% reduction in household salt consumption and a 30% reduction in the salt content of bread and biscuits was achieved, the range of iodine intake from these foods would be from 63 µg in North Western province to 101 µg in Eastern province. The respective percent of iodine reference values for adults would range from 66% to 106% of the EAR, 42% to 67% of the RNI.

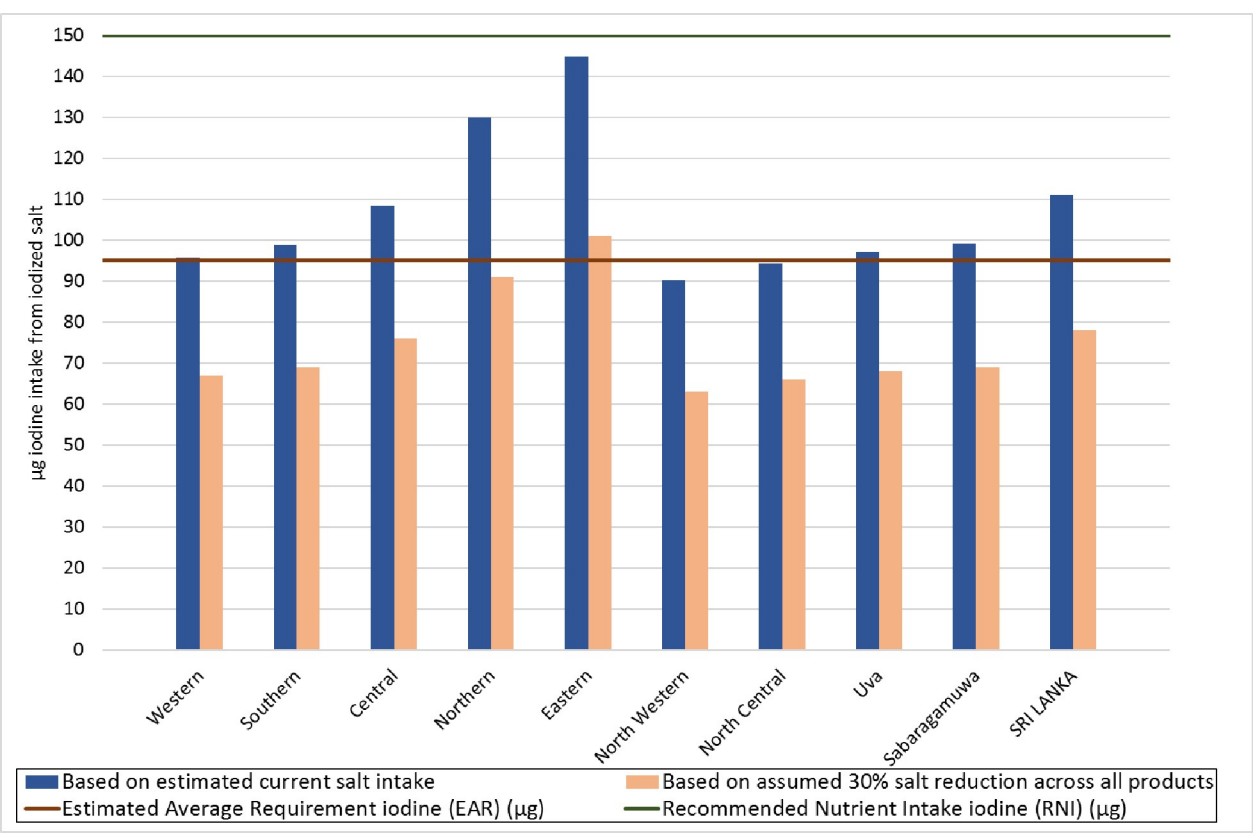

**Fig 2. Potential impact of successful 30% salt reduction on iodine intake from household salt and salt in selected IPF in relation to iodine intake reference values by provinces.** [Tolerable Upper Limit for iodine intake = 600 μg not shown].

## Discussion

Evidence from this assessment shows that of total salt intake from household salt and the 3 selected IPF in Sri Lanka, the selected IPF contribute to around 20 percent of this salt intake. Household salt remains by far the major source of salt (and iodine from this salt) in the diet. The modelling conducted for this paper indicates that typical current estimated iodine intake from iodized household salt and food industry salt is slightly above the EAR for iodine, which is the level of intake estimated to meet the requirements of half of healthy non-pregnant adults in the population. Yet this level of intake would only contribute to three quarters of the RNI for iodine, which is the average intake considered sufficient to meet the nutrient requirements of nearly all (97–98%) of healthy non-pregnant adults in the population [15]. However, despite iodine intake from these sources of iodized salt seeming insufficient to meet the needs of all population groups, particularly in North Western and North Central provinces where intake estimates donot meet the EAR for iodine; survey evidence indicates optimal iodine status among school age children in all provinces and among pregnant women at the national level [5, 7]. This suggests that there are other sources of iodine in the diet, for example, iodized salt in other processed foods and foods made outside the home that were not included in this assessment. It is also known that iodine in drinking water contributes to intake in some areas of the country [5]. Based on the demonstrated significant current iodine intake from house-hold and food industry iodized salt and on data showing adequate iodine nutrition among the population, the current salt iodine standards (15–30 mg/kg) appear adequate to meet dietary

needs and sustain optimal iodine intake among all age groups [5]. The low percentage (< 30%) of the UL for iodine met by intake from iodized salt in household salt and the selected IPF indicates that current and potential iodine intake from iodized salt does not pose any risk of adverse health effects to adults.

National iodine surveys and food industry data indicate that most food grade salt is adequately iodized [5, 6], except where exemptions to its use apply, as is the case for dried fish. It will be important to maintain this high coverage of adequately iodized salt at household level as this does appear to be a major source of iodine in the Sri Lankan diet. The assessment reported in this paper was very timely, considering that the Sri Lankan diet appears to be in transition from a reliance on almost entirely home prepared foods to increased consumption of IPF and other foods bought outside the home. Experience from other countries indicates this transition is often driven by factors that include an increase in the number of women working outside the home, greater availability and convenience of IPF, and their effective advertising, and marketing [17, 18].

A finding from this paper is that overall salt consumption in Sri Lanka is high, averaging approximately twice the maximum WHO recommended level for salt intake for adults [1]. The main sources identified in this assessment were household salt and dried fish; intake of salt from both these sources will be more difficult to address through the national salt reduction strategy than, for example, reformulation of IPF such as bread and salty snacks to have lower salt content. Nevertheless, evidence on the risks associated with current levels of salt intake suggests it is imperative for the Ministry of Health to make all efforts to achieve the target of overall 30% reduction in population salt intake by 2025 [2]. A potential consequence of achieving the salt reduction target is a decrease in iodine intake from iodized salt, to below the EAR. The risk of this to population iodine status could, however, be assessed through careful monitoring of population iodine status alongside strengthened regulatory monitoring and enforcement of the use of iodized salt by the food industry, to ensure the continued high level of engagement and compliance. Where data from these suggested monitoring activities indicates that iodine intake is below optimal, the salt iodine standard could be adjusted to compensate. The current exception to food industry use of iodized salt is in the production of dried fish industry, which has a government exemption [4].

Strengths of the assessment reported in this paper were the availability of consumption data for household salt and bread at national and provincial level and for dried fish and biscuits at national level, cooperation from the food industry in providing estimates for salt content of bread and biscuits, and the availability of laboratory data on the salt content of commonly consumed dried fish. Limitations of this assessment related to the reliability of the data it was based on. This included the fact that consumption estimates were based on average quantity per household member, regardless of age, since information to convert household intake using adult male equivalence was not available. The resulting data may not have been fully representative of typical intake quantities among non-pregnant adults, which was the basis used for comparative iodine intake reference values. There was also an assumption that all salt used by bakeries and biscuit manufacturers was iodized in accordance with legislation, this was supported by information from the respective industries but was not verified. Additionally, the lack of provincial data for biscuit and dried fish consumption meant that an assumption was made that the consumption level for biscuits and dried fish was the same throughout the country, which may underestimate or overestimate the results at provincial level. Finally, the assessment of possible impact of salt reduction on iodine intake applied a 30% reduction across all products, whereas in reality, it is likely that some products would be more appropriate to target for reformulation to reduce salt content than others.

## Conclusion

This study indicates an overall well-functioning salt iodization program in Sri Lanka that is contributing to sustained adequate iodine status of the population. There is a remarkable contribution from the private sector (salt industry and food industry) to make salt iodization and elimination of iodine deficiency disorders a success. The IGN programme guidance provided a useful tool to assess IPF consumption, and the relative salt intake and iodine intake from selected foods, along with household salt, among the population. A successful salt reduction initiative will reduce iodine intake, however, careful monitoring of iodine status and continued regulatory monitoring and enforcement of food grade salt iodization, will help determine whether the impact on iodine status will require adjustments to the national salt iodization strategy, including to salt iodine standards.

The purpose of this study was to present an illustration of different major sources of dietary salt, and the potential for these to contribute to iodine intake if the salt was iodized that could be helpful for strategy development and to establish a template for future assessments with new data as they become available. Despite some limitations, our study achieves this and highlights the importance of combining implementation and monitoring of the salt reduction strategy to prevent chronic diseases with implementation and monitoring of the salt iodization strategy to eliminate iodine deficiency disorders. This combined implementation is recommended by WHO to ensure that the approaches do not conflict with each other. It is useful to identify the contribution of IPF to salt intake in implementing both policies. Considering the high level of salt intake, monitoring of salt content in IPFs along with health promoting actions will be necessary to ensure industry compliance and population engagement to achieve these goals.

## Acknowledgments

We thank staff of Department of Nutrition, Medical Research Institute, Ministry of Health, Sri Lanka for providing laboratory data to support this analysis and Ceylon Biscuit Limited, Maliban Biscuit manufacturers, All Ceylon Bakery Association, for providing all necessary data. We would like to thank Dr Robin Houston, senior advisor, IGN for supporting in every step.

## Author Contributions

**Conceptualization:** Renuka Jayatissa, Jacky Knowles, Karen Codling, Jonathan Gorstein.

**Data curation:** Renuka Jayatissa, Chandima Haturusinghe, Karen Codling.

**Formal analysis:** Renuka Jayatissa, Chandima Haturusinghe, Karen Codling.

**Funding acquisition:** Renuka Jayatissa.

**Investigation:** Renuka Jayatissa, Chandima Haturusinghe.

**Methodology:** Renuka Jayatissa, Chandima Haturusinghe, Jacky Knowles, Karen Codling.

**Project administration:** Renuka Jayatissa, Chandima Haturusinghe, Jonathan Gorstein.

**Resources:** Renuka Jayatissa.

**Supervision:** Jacky Knowles.

**Visualization:** Chandima Haturusinghe.

**Writing – original draft:** Renuka Jayatissa, Chandima Haturusinghe, Jacky Knowles, Karen Codling, Jonathan Gorstein.

**Writing – review & editing:** Renuka Jayatissa, Chandima Haturusinghe, Jacky Knowles, Karen Codling.

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
