## [Decision Letter · Decision Letter 0]

1 Jun 2021

PONE-D-20-39099

Contribution of processed foods to salt intake and iodine intake of Sri Lankan population and the potential impact of salt reduction on iodine intake

PLOS ONE

Dear Dr. Jayatissa,

Thank you for submitting your manuscript to PLOS ONE. After careful consideration, we feel that it has merit but does not fully meet PLOS ONE’s publication criteria as it currently stands. Therefore, we invite you to submit a revised version of the manuscript that addresses the points raised during the review process.

The authors limited the analysis to only three industrially processed foods (bread, dried fish and biscuits). Other products like canned fish, margarine, cheese and butter were excluded based on the assumption that they were not frequently consumed. Further other important sources of dietary salt including tomato ketchup has not been mentioned. I recommend the authors to do sensitivity analysis on this decision of excluding these foods from the analysis.The study estimated average daily Individual consumption of salt and industrially processed food by dividing food available at household level by number of household members. The estimation seems erroneous because it assumed that all household members consumed equal amount of food. I recommend the authors to take the age and sex profiles of the household members into consideration and recalculate adult-equivalent level of consumption for the households.How did the authors combined district level consumption data and estimate daily per capita consumption for each province? Assuming the population size of the districts is different, weighted analysis should be applied.

We look forward to receiving your revised manuscript.

Kind regards,

Samson Gebremedhin, PhD

Academic Editor

PLOS ONE

Additional Editor Comments (section-by-section comments):

Introduction

Is there any empirical evidence that there is shifting towards increased consumption of IPF in Sri Lanka? In the introduction section, can you please present numeric figures supporting this shift?

Methods

Line 100-1: “The survey also collected information on the quantity of salt purchased at household level monthly”. What if salt is not purchased in that schedule? How did you handle such scenario?

Line 133: “The average percent salt content of dried fish from the laboratory analysis was 16%.” Have you tried to validate this lab finding with existing literature?

Results

Line 157-61: Rather than taking the highest estimate of 8.5mg, why it was not possible to take mean of means?

Discussion

Please discuss the implication of assuming equal intake of dried fish and biscuits across all the provinces.

Journal Requirements:

2. In your Methods section, please provide additional information about 1) the data sources used (specifying what variables were considered, how these were defined and categorised, etc); 2) the model applied ( for example, the equations representing the model; how the model was calibrated;  what parameters  and assumptions were applied.

[This research received grant from IGN.]

 [Initials of the authors who received each award - RJ

Grant numbers awarded to each author -

Investment ID OPP1195090

The full name of each funder -

Bill and Melinda Gate Foundation

URL of each funder website - None

Did the sponsors or funders play any role in the study design, data collection and analysis, decision to publish, or preparation of the manuscript?

No

The funders had no role in study design, data collection and analysis, decision to publish, or preparation of the manuscript.]

4. We noted in your submission details that a portion of your manuscript may have been presented or published elsewhere.

[No, only presented to ministry of health for policy advocay]

Reviewers' comments:

Reviewer's Responses to Questions

**Comments to the Author**

1. Is the manuscript technically sound, and do the data support the conclusions?

Reviewer #1: Partly

Reviewer #2: Yes

2. Has the statistical analysis been performed appropriately and rigorously? 

Reviewer #1: No

Reviewer #2: Yes

3. Have the authors made all data underlying the findings in their manuscript fully available?

Reviewer #1: No

Reviewer #2: Yes

4. Is the manuscript presented in an intelligible fashion and written in standard English?

Reviewer #1: Yes

Reviewer #2: Yes

5. Review Comments to the Author

Reviewer #1: The manuscript entitled “Contribution of processed foods to salt intake and iodine intake of Sri Lankan population and the potential impact of salt reduction on iodine intake” (PONE-D-20-39099) had been reviewed and the comments are listed below.

Presented manuscript is interesting and the estimation of the contribution of industrially processed food to salt and iodine intake is now a frequent topic of debate among nutritionists and the food industry. Nevertheless to clear up all given information, major supplementations need to be done. Consequently the authors are asked to respond to these comments.

Manuscript’s title should be corrected as it does not fully reflect to the content since the intake of salt and iodine was estimated on a very narrow group of products.

The term of industrially processed food should be defined. The Authors included dried fish, bread and biscuits. Could it be named a industrially processed food? Reading the paper one has a feeling that the meaning would be to reduce the salt consumption on ultra-processed food, but actually there is no clear information weather foods selected for estimation were of industrial or homemade. When analyzing the iodized salt intake it should not be forgotten, that iodine components are susceptible to numerous changes e.g. sunlight, oxygen, temperature, presence of other food components, which eventually can significantly deplete the iodine content in the product. There are no information about the potential changes during food preparation and processing conditions, as well as iodized salt storage time. All those factors significantly influence the iodine intake.

It is not claer what is the average consumption of the analysed food. Used for the research HIES quantitative data are from 2016, the Department of Census and Statistics decided to conduct the HIES once in every three years in Sri Lanka, which shows the data may not be up to date. Is that the most consumed food by all consumers, so its salt and iodine content would whether represent the surveyed group of consumers, or is it consumed occasionally, also due to the changing nutritional trends of Sri Lanka citizens? Needs clarification.

The Material and method section is too long and not informative, the modelling process used for the paper preparation is not clear. The section needs ordering.

Analysis of the material in the publication indicates that the study may not be representative of salt and iodine intake as it only covers 3 groups of products: bread, biscuits and dried fish. The authors presented the limitations of their research (line 279-291), but their importance may indicate too many significant ambiguities and shortcomings of the analysis performed.

The Authors did not include a statistic analysis information and how the modelling process was conducted. If the results are estimated as described in the tables it is not sufficient for a scientific paper and needs to be supplemented.

Above mentioned comments confirm need of careful corrections and supplementations to obtain proper scientific load. Therefore, I do not recommend it in its current form for further steps of PLOS ONE publication process. My recommendation is major revision, since, as described above, the work requires proper suplementation.

Reviewer #2: It should be better specified whether potential iodine intake from household salt and IPF salt sources was calculated assuming that salt was iodized to 15 mg/kg or 30 mg/kg or to the mean value of 21 mg/kg cited at line 56. In Fig.1 it is reported that estimated current iodine intake is based on 78% of household salt being iodized to at least 15 mg/kg, but in the Method section a range of 15-30 mg/kg is indicated (line 143). This point should be clarified.

In the abstract and through the text the expression "Industrially Processed Foods" is abbreviated as IPF and IPFs. Please use one of them.

At lines 146-147 the Authors state “.....78% of housholds were using iodized salt with some amount

of iodine while 78% of households use salt with at least 15 mg/kg iodine”. The first percentage is probably higher than 78%. Is it 95% as stated at line 54?

6. PLOS authors have the option to publish the peer review history of their article (what does this mean?). If published, this will include your full peer review and any attached files.

Reviewer #1: No

Reviewer #2: No

---

## [Author Response · Author response to Decision Letter 0]

15 Jul 2021

Title was revised as suggested. Additional text has been added to address this comment. Including clarification of the assumption, for the purpose of the modelled illustration, that consumption data for the selected foods related only to industrially produced foods. 

A loss of 30% iodine from salt at production to final consumed product was already included in the methods and calculation, however, we have further clarified why 30% was used. (Lines 186 to 192 and 209-211)

The latest available HIES data for Sri Lanka is still the 2016 survey according to the national statistics office website.

The foods were selected as those being most frequently consumed by women of reproductive age and the highest consumption quantities in the HIES. Indicating these foods are consistently widely consumed. 

In addition, selection of processed foods was based on those with consumption levels and salt content that would result in an expected daily intake of at least 1.5 µg iodine. This is now more clearly explained (lines 114 to 117 and 141 to 148) 

The study was a preliminary assessment for strategic review of foods contributing to iodine intake through iodised salt, partly to address some concerns about the suitability of salt iodine levels. The discussion and limitations make it clear that household salt is likely still the major contributor to salt/iodized salt intake, however it will be important to consistently review changes in consumption of IPF and re-run the analysis as consumption of other salt containing IPF increase.

The methods in the IGN programme guidance and those applied in this study are not intended as rigorous statistical research but as an evidence base for a public nutrition strategy. Additional information about the equations used has been added to the methods to make that part of the assessment clear.

It was presented to PLOS as a demonstration of a public nutrition intervention to guide strategy decisions, not as rigorous scientific research. It was accepted for submission as part of a collection of a methodological overview (which you should have received along with this manuscript) and national experiences of implementing the process from 3 other countries. The overview paper compares and contrasts the findings from each, in terms of public nutrition strategy.

---

## [Decision Letter · Decision Letter 1]

3 Sep 2021

Estimated contribution of most commonly consumed industrialized processed foods to salt and iodine intakes in Sri Lanka

PONE-D-20-39099R1

Dear Dr. Jayatissa,

We’re pleased to inform you that your manuscript has been judged scientifically suitable for publication and will be formally accepted for publication once it meets all outstanding technical requirements.

Kind regards,

Samson Gebremedhin, PhD

Academic Editor

PLOS ONE

Additional Editor Comments (optional):

Reviewers' comments:

Reviewer's Responses to Questions

**Comments to the Author**

1. If the authors have adequately addressed your comments raised in a previous round of review and you feel that this manuscript is now acceptable for publication, you may indicate that here to bypass the “Comments to the Author” section, enter your conflict of interest statement in the “Confidential to Editor” section, and submit your "Accept" recommendation.

Reviewer #1: All comments have been addressed

Reviewer #2: All comments have been addressed

2. Is the manuscript technically sound, and do the data support the conclusions?

Reviewer #1: Yes

Reviewer #2: (No Response)

3. Has the statistical analysis been performed appropriately and rigorously? 

Reviewer #1: N/A

Reviewer #2: (No Response)

4. Have the authors made all data underlying the findings in their manuscript fully available?

Reviewer #1: Yes

Reviewer #2: (No Response)

5. Is the manuscript presented in an intelligible fashion and written in standard English?

Reviewer #1: Yes

Reviewer #2: (No Response)

6. Review Comments to the Author

Reviewer #1: Considering the submitted revision of the manuscript, the replies given and the changes made by the authors corresponding to each concern raised by the reviewer are satisfactory and convincing, therefore the quality of the manuscript has improved significantly. For these reasons I recommend acceptance of the work in its present form and publication in PLOS ONE journal.

Reviewer #2: (No Response)

7. PLOS authors have the option to publish the peer review history of their article (what does this mean?). If published, this will include your full peer review and any attached files.

Reviewer #1: No

Reviewer #2: No

---

## [Editor Report · Acceptance letter]

10 Sep 2021

PONE-D-20-39099R1 

Estimated contribution of most commonly consumed industrialized processed foods to salt intake and iodine intakes in Sri Lanka 

Dear Dr. Jayatissa:

I'm pleased to inform you that your manuscript has been deemed suitable for publication in PLOS ONE. Congratulations! Your manuscript is now with our production department. 

Kind regards, 

on behalf of

Dr. Samson Gebremedhin 

Academic Editor

PLOS ONE